# Comprehensive Recovery of Metals in Tailings Utilization with Mechanochemical Activation

Vladimir I. Golik [1,2], Mikhail F. Mitsik [3], Yulia V. Aleksakhina [4], Elena E. Alenina [4], Natalia V. Ruban-Lazareva [5,6], Galina V. Kruzhkova [7], Olga A. Kondratyeva [7], Ekaterina V. Trushina [7], Oleg O. Skryabin [8,9] and Marat M. Khayrutdinov [10,*]

1   Metallurgy Department, Moscow Polytechnic University, Bolshaya Semyonovskaya Str., 38, 107023 Moscow, Russia; v.i.golik@mail.ru
2   Mining Department, North Caucasian Institute of Mining and Metallurgy, State Technological University, Nikolaeva Str., 44, 362021 Vladikavkaz, Russia
3   Department of Mathematics and Applied Information Science, Institute of Service and Entrepreneurship (Branch) of DSTU, Shevchenko Str., 147, 346527 Shakhty, Russia; m_mits@mail.ru
4   Management Department, Moscow Polytechnic University, Bolshaya Semyonovskaya Str., 38, 107023 Moscow, Russia; aleksahina@yandex.ru (Y.V.A.); alenina@mail.ru (E.E.A.)
5   Department of State and Municipal Finance, Plekhanov Russian University of Economics, Stremyanny Lane, 36, 117997 Moscow, Russia; rubanlazareva@mail.ru
6   Department of Taxes and Tax Administration, Financial University under the Government of the Russian Federation, Leningradsky Prospect, 49, 125167 Moscow, Russia
7   Economics Department, National University of Science & Technology (MISIS), Leninsky Ave., 4., 119049 Moscow, Russia; galkruzhkova@mail.ru (G.V.K.); kondrateva.oa@misis.ru (O.A.K.); trushina.ev@misis.ru (E.V.T.)
8   Department of Industrial Management, National University of Science & Technology (MISIS), Leninsky Ave., 4., 119049 Moscow, Russia; 88-88@mail.ru
9   Department of International Economic and Financial Relations, Russian State Academy of Intellectual Property, Miklukho-Maklaya Str., 55a, 117279 Moscow, Russia
10  Itasca Consultants GmbH, Leithestrasse Str., 111a, 45886 Gelsenkirchen, Germany
*   Correspondence: profmarat@gmail.com

**Abstract:** The paper analyzes the results of metal extraction from tailings of ore processing based on traditional approaches. The history of methods of secondary processing of mineral raw materials is described. The technique and features of mechanochemical activation of the leaching process of metal ores and coals are described. The results of laboratory studies of a new mechanochemical technology for metal extraction are presented. A description of the compromise optimal criterion formulated in the model of extraction of metals from tailings with substandard mineral raw materials based on the mathematical planning of the experiment, regression analysis methods and consideration of sanitary standards for the disposal of processing tailings is proposed. Mechanochemical technology results in economic and environmental effects from the utilization of processing tailings and radical strengthening of the mineral resource base of the mining industry.

**Keywords:** deposit development; tailings; ore processing; leaching; metals; utilization; economy; ecology; mechanoactivation; mechanochemistry

## 1. Introduction

Increasing the efficiency of metal extraction from processing wastes [1,2], metallurgical slags [3,4] or wastewater from mines and concentrating plants [5] is closely related to environmental [6] and economic [7] problems. The solution to these problems is associated with aspects of the transformation of mining and processing waste by reducing the heavy metal content to zero [8]. The priority view is the need to reduce the global hazard of the impact of ore tailings on the environment [9], which can only be achieved by the complete utilization of environmentally hazardous tailings [10]. Other approaches, for example,

biological waste reclamation without chemical extraction of hazardous components [11], not only have lower efficiency but also create a danger of migration and accumulation of chemically harmful mobile substances in the soil [12,13].

The preservation of the Earth's pristine nature and mineral diversity is often proposed by transferring mining production to space bodies [14]. However, the lack of technical possibilities and unresolved legal problems [15,16] postpone this idea for the distant future.

The main challenge in tailings processing is to minimize (up to complete exclusion) the leakage of harmful elements into the environment because the tailings themselves remain in the natural environment [17]. The dependence of the ecological and economic efficiency of tailings processing with the neutralization of environmentally hazardous elements has the form:

$$P_t > P + C,$$

where $P_t$—total profit from tailings neutralization; $P$—penalties for environmental damage; $C$—total tailings processing costs.

Currently, the neutralization of harmful components in tailings is not given much attention in Russia due to the following reasons [18]:

- Land rents or environmental fees are commensurate with and often below the profits obtained from the additional production;
- Damage to the health of employees or residents of the mining region is borne by the state (unemployment benefits; chronic diseases, disability, etc.).

The involvement of industrial waste in a closed production cycle for producing backfill [19], manufactures of civil engineering [20,21], from an ecological point of view, is an effective method of waste disposal [22].

Tailings of ore processing pollute the environment during their storage [8,23], which creates environmental risks [17,24]. Consequently, the neutralization of tailings can be environmentally and economically efficient, provided that the processing method is optimal. A new approach to ore processing technology involves a non-traditional method of stripping the material and the complex extraction of metals into the agent solution [25].

At present, the situation with tailings neutralization is such that conventional technologies do not exclude the chemical leakage of metals into the processing waste while recovering metals by mechanical energy alone. The process of the modernization of tailings processing is realized on the basis of a set of chemical and hydrometallurgical measures, which combine different types of energy [26].

Current approaches to assessing the damage to humans, flora and fauna from the storage of processing tailings strive to eliminate the tailings themselves, as it is very difficult to reliably assess the harm from their storage. Efforts have therefore focused on the disposal of tailings and the neutralization of their hazardous impact on nature.

The reason for not completely extracting all metals in conventional ore processing methods is that only two methods are used to extract valuable components. These methods are designed for a limited set of raw material components. Metal leaching technologies have been proposed for the utilization of processing tailings. However, chemical leaching alone takes a long time, even if mechanical activation is used, is not economically feasible and does not provide the full recovery of valuable components.

The developing mechanochemical technology of tailings processing differs from traditional technologies by the fact that the extraction of metals in the disintegrator is accelerated with the energy arising from the increased rotor velocity and particle impact. The novelty of the effect is that the leaching solution is not filtered through the cracks, but is pressed into them, and the extraction of metals occurs during the destruction of crystals [25].

The ore treatment methods used are characterized by a loss of components in tailings (Figure 1).

However, the simultaneous presence of valuable and harmful components in industrial wastes prevents their wide use in a closed, waste-free (low-waste) production cycle. Therefore, the extraction of components contained in industrial wastes to a zero

level and the subsequent environmentally friendly use of waste in the construction and road industries is the most effective method of utilization. This recycling scheme will have a multiplicative effect: economic (profits from the extracted valuable components); environmental (avoidance of the possibility of harmful components migration).

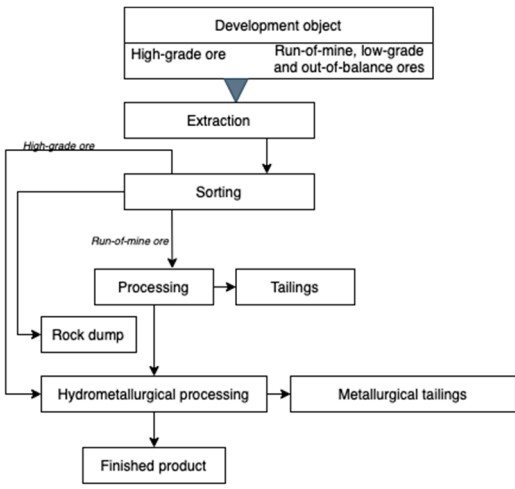

**Figure 1.** Conventional metallic ore processing scheme.

Based on the above, it can be concluded that the development and implementation of methods to extract valuable components from industrial waste is a highly topical task.

## 2. Objects and Methods

A new approach to the technology of the processing of refractory ores and their concentrates includes unconventional stripping methods and the complex recovery of metals in agent solution [27].

Historically, the industrial scale of metal leaching was mastered during World War I in the United States, Japan, and South American countries, amongst others. The heap leaching of metals is applied in cases of necessity to neutralize ore sorting tailings and their processing. For example, the results of the high-efficiency leaching of gold from processing tailings with a content of 0.3. . .0.6 g/t are known in world practice.

The Manybai uranium deposit in Northern Kazakhstan has been developed since the late 1980s using acidic solutions to process uranium ore tailings with a volume of 1.5 million tons (Figure 2) [28].

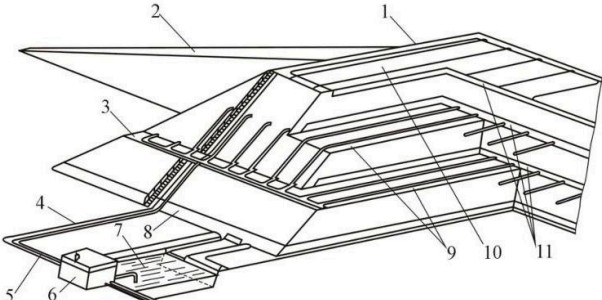

**Figure 2.** Leaching of uranium and gold in a stack: 1 stack; 2 pass; 3 bank; 4 mortar pipeline; 5 air duct; 6 pumping station; 7 sump; 8 waterproofing; 9 aeration system; 10 irrigation; 11 fine material.

For the purpose of intensification, leaching processes were carried out with a variable electric field with a frequency of 5 to 40 Hz.

The largest gold extraction plant in the CIS, Kyzylkumredmetzoloto Concern (Uzbekistan), uses gold leaching technology for Muruntau open pit ores. Earlier studies proved

the prospect of the cumulative leaching of heavy metals [29], but some of the valuable components remain in the tailings and fall into the soil, contaminating it [30,31].

The intensification of leaching processes using the activation phenomenon makes it possible to increase the recovery of metals such as gold, copper and uranium to certain degrees.

Over the last two hundred years, leaching theory has dealt with pressure, temperature, the dispersion of substances (mechanical or chemical) and catalysis. Modern technological processes already use a fundamentally new phenomenon where the state of a mineral is changed due to the application of high-mechanical energy to the substance [32].

Metal leaching is promising, but it does not provide the right process parameters, as it takes a long time for the agents to penetrate deep into the mineral particles. The most interesting technologies are those that combine the capabilities of chemical processing and disintegrators.

Since the second half of the twentieth century, new technologies have been created for recovering metals from ores. One of the main methods is the mechanical activation of matter by applying a large amount of mechanical energy to the ore crystals. Processing the ore crystals in a disintegrator causes the ore particles to break apart due to rapid successive impacts. The impact velocity reaches up to 250 m/s. As the temperature of the material being crushed rises, the structural state of the material changes. For the crushing particles, there are loads four hundred million times higher than the acceleration of gravity [2,10,20,33].

Previously, the term "mechanical activation" was understood as an increase in the catalytic properties of a substance when it was subjected to grinding by a vibratory mill [34]. This results in an acceleration of the chemical reaction [35] and an increase in the compressive strength of the materials obtained from the processing tails [33]. The device for tailings activation is a disintegrator, which consists of two rotors enclosed in a common casing, mounted on two coaxial shafts and rotating in opposite directions (Figure 3).

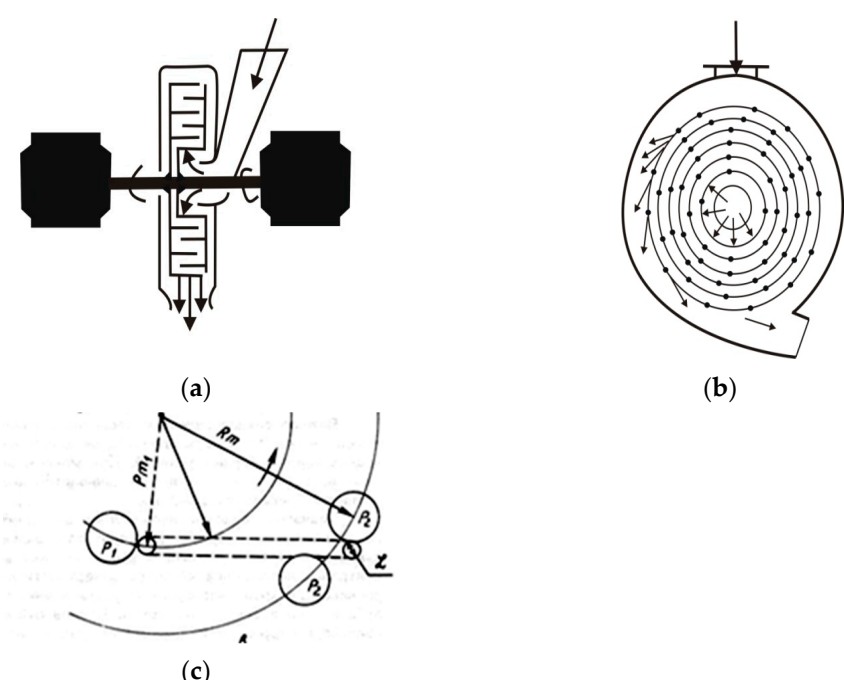

(a)

(b)

(c)

**Figure 3.** Principle of the disintegrator: (**a**) Device view; (**b**) Force distribution in the field of the working body; (**c**) Movement pattern of the crushing particle.

The disintegrator rotor disks are fitted with circular cylindrical fingers equally spaced on the rotor's axis of rotation, the number of fingers being two to four and arranged at equal angles of rotation: 1800, 1200 or 900, depending on the number of fingers.

The tailings are fed into the center of the rotor, where gravity and centrifugal forces act on the feed material, causing it to rotate towards the inner boundary of the disintegrator and the material is crushed under finger blows at a high linear velocity and rotation frequency. The linear velocities are doubled by their rotation with the opposite orientation. During the disintegration process, the material accumulates thermal energy, the values of which can be up to 30% (e.g., for $SiO_2$) of the energy used to disintegrate the material.

The effect created in the disintegrator is commonly referred to as high-mechanical energy activation. Disintegration impact velocities are higher than in ball mills or vibratory mills, with the acceleration between particles millions of times greater than gravity.

The activation process produces a large set of electrically irregularly charged centers in the crushed material: much larger than in the case of other mechanical impacts. The material particles break down along the boundaries of the impurity accumulation. In the case of the disintegration of polycrystalline material, breaking occurs along the surface of crystal particles, which, in turn, entails the breaking of material phases with lower strength, including breaks along the boundaries of phase boundaries. As a consequence, the processes of separation of each phase of the crushed material in the disintegrator are significantly accelerated, which increases the yield of the desired product.

For the first time in world practice in the mining industry, the disintegrator was applied during the development of the Shokpak deposit in mine №4 of the Tselinny mining and chemical combine (Northern Kazakhstan) under the guidance of the author of this article [26,28] (Figure 4).

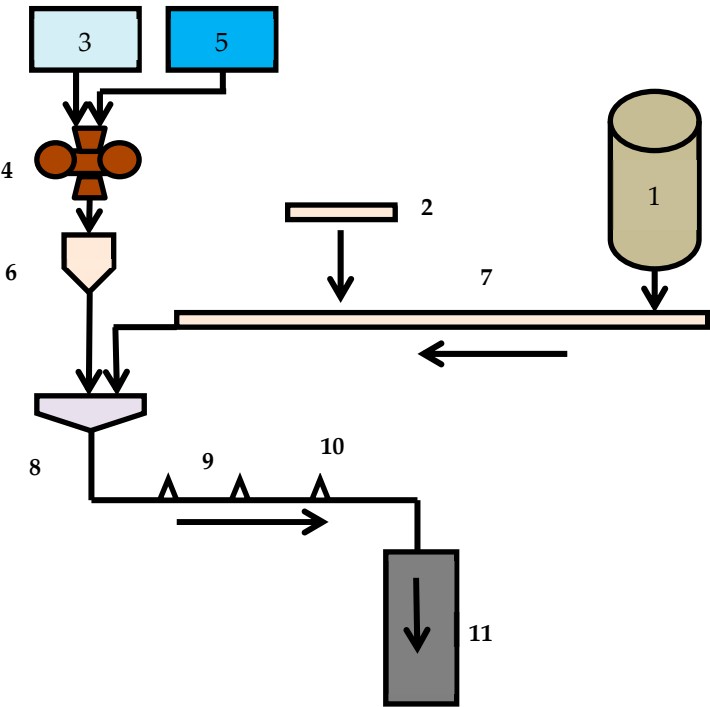

**Figure 4.** The stowing complex with activation of hardening mixture components: 1 cement hopper; 2 aggregate vibrating screen; 3 blast furnace slag; 4 disintegrator; 5 activated water; 6 vibrating mill; 7 conveyor; 8 mixer; 9 stowing pipeline; 10 vibrators; 11 block chamber.

The development of leaching technology lies in the fact that, with the use of a disintegrator, the leaching of metals simultaneously occurs with the destruction of crystals, and the leached solution is pressed into cracks that occur during the crushing of the material.

The leaching of the metal into the solution occurs more efficiently during the disintegration of polycrystalline raw materials due to the increase in the contact area of the catalytic substances with the metal particles, which makes it possible to increase the leaching rate and the percentage of the transition of metals into the solution.

In general, the process of leaching metals into the solution can be described in the form of one-dimensional equations of motion continuity:

$$\begin{cases} \frac{\partial a}{\partial \tau} + \frac{\partial c}{\partial \tau} + k\frac{\partial a}{\partial x} = 0, \\ a = f(c) \end{cases}$$
(1)

where, $a$—total capacity of sorbents, mg/g; $c$—concentration of metals in solution, mg/L; $\tau$—leaching duration, h; $k$—ratio of the residence time of ore crystals in the leaching process to the application time of the sorbents; $x$—vertical coordinate in the leach tank; $a = f(c)$—leaching isotherm equation.

Let the concentration of metals in the solution at the initial moment be:

$$a(x,0) = a_0,$$
(2)

The complexity of the numerical solution of problems (1) and (2) is caused by the presence of different metals in the solution, different leaching rates as well as the heterogeneity of metal distribution in ore crystals. In this regard, the efficiency of metal extraction is proposed to be investigated by the mathematical planning of experiments.

Such an effect arises for two independent reasons: the surface of the material fractions is opened up to a particle fineness of 0.074 mm and the forces of intermolecular interaction in the material are reduced.

It should be noted that tailing crushing processes in the disintegrator, placed as a transmission link between the mill and the mixer, take place in both dry and liquid crushing modes.

Tailings are not used to form hardening mixtures if they contain metals and sulfur. Reagents are delivered to the disintegrator during tailings crushing, which increases the share of recoverable metals, sulfur and other valuable components by up to 50–70% and allows for the creation marketable products.

The disintegrator crushing results in a small proportion of fractions larger than 125–400 μm and fractions smaller than 5 μm, while the bulk of the fractions are between 5 μm and 125 μm. When grinding in a ball mill, highly dispersed fractions provide a significant specific surface area (Ssp = 4930 cm$^2$/g) with a relatively small proportion of its content (10.6%). The indicators of the specific surfaces of slags can differ by one and a half times during single and double grinding, and the proportion of fractions with a size of no more than 63 μm during double crushing is significantly higher [2,25–28].

The approach to the deep processing of tailings is based on combining the chemical extraction of valuable components with tailings activation inside the working body, resulting in the extraction of metals into the solution during the crushing of the ore.

Under certain conditions, oxidizing reagents remove almost all of the metals in the tailings into solution by up to two orders of magnitude faster. The difference between mechanochemical technology and conventional technology is the ability to extract valuable components from recycled raw materials to a level comparable with sanitary requirements, so that secondary tailings can be used without harming the environment.

The principle of disintegration has been known for around 150 years. The phenomenon of leaching has been developed since the middle of the last century. The idea of leaching in a disintegrator belongs to the authors of this article [27].

In previous studies, special attention has been paid to the processes of mechanochemical activation and the improvement of its methodology. The use of mechanical activation during leaching allows for the extraction of a valuable component in the working body of the disintegrator. This method speeds up the leaching process by two orders of magnitude and increases the share of recoverable valuable components by one and a half to two times.

The relevance of tailings neutralization lies in the preference for environmentally friendly technologies in underground mining where tailings can be the raw material for hardening mixtures.

The use of tailings without metal extraction for the production of backfill has been known in mining practice [19].

The feasibility of the mechanochemical leaching of metals from mineral tailings was carried out by the authors for non-ferrous and ferrous metallurgy waste and coals of Russian regions using the DU-11 disintegrator made at the "Applied Mechanochemistry "Gefest" Centre.

The aim of the experiments was to compare traditional agitation leaching into the solution and the new accelerated leaching in a disintegrator.

The tailings leaching variants that were compared:

1. Agitation leaching;
2. Agitation leaching after activation in a disintegrator;
3. Leaching in a disintegrator;
4. Agitation leaching and leaching in a disintegrator;
5. Repeatedly leaching in the disintegrator.

Considering the spiral shape of the disintegrator working body, the description of the leaching process based on a two-dimensional Cauchy problem in polar coordinates would be more accurate. The leaching parameters would be determined using a finite-difference scheme, where it would be necessary to determine the stability or prove the theorem of the existence and uniqueness of the solution. The presence of different metals, their varying leaching rates and the heterogeneous distribution of metals in the ore complicate the task. Therefore, a simpler approach to the problem of determining the parameters of metal leaching into solution was chosen, which allows for describing the leaching process with satisfactory accuracy.

The experiments were carried out with mathematical scheduling according to the Wenken-Box plan. This approach proposes an active experiment with controllable and independent input parameters. The requirement of the controllability of factors means that all combinations of factors are feasible, while the independence of factors means that any factor at each level can be accepted regardless of the values of the other factors.

The main problem in the experiment planning is the choice of the area of variation, since this task is unformalizable. The boundaries of factor variation were chosen according to the fulfillment of sanitary requirements for secondary tailings processing, as well as finding a metal recovery solution close to a compromise optimum.

Independent factors in research planning and boundaries of factor variation in the experimental plan based on stated requirements:

X1—factor of $H_2SO_4$ content in the leaching solution, varies from 2 to 10 g/L;

X2—factor of NaCl content in the leaching solution, varies from 20 to160 g/L;

X3—factor of the weight ratio of the leaching solution to the amount of leaching mass in a single experiment, varies from 4 to 10;

X4—leaching time factor, varies from 0.15 to 1.0 h.

The volume of tailings that were leached in a single experiment of the first, third, fourth and fifth stages was 50 g.

In the second stage experiments, the number of tailings was increased to ensure that the volume of leach tails was also 50 g after the fine fractions had been carried away by the dust.

The leaching solution was pre-prepared according to the defined composition and then mixed with the processing tailings. For the first stage, the solution was mixed before agitation leaching. For the second stage, the solution was mixed with tailings after activation of the leaching material in the disintegrator. For the third, fourth and fifth stages, it was mixed before loading into the disintegrator.

The first and second stages are characterized by agitation leaching of the tailings in pulp solution. This pulp was obtained by adding activating material to the leaching solution, or by passing tailings in solution through the disintegrator. The leaching was performed for a given period of time in the agitator at a constant angular velocity.

The third stage is that the mixture obtained by adding tailings or ore to the leaching solution is fed into the disintegrator chamber.

The fourth stage is characterized by the presence of the mixture in the working chamber of the disintegrator, then leaching for a defined period using the agitator at a constant angular velocity.

The fifth stage consisted of passing the leaching solution with the materials through the working zone of the disintegrator three to four times, with an increase in the activation time, but with an increase in the proportion of valuable components extracted into the solution.

At the end of each leaching stage, the resulting pulp was examined for qualitative composition.

Five batches of tailings were taken for each experiment using independent and equal material sampling. For each type of tailings, 0.05 t of processing tailings seeded through a 2.0 mm grid were prepared. This approach ensures that the reliability of modeling results according to the experimental plan is 95%, at a significance level of 0.05.

The chemical composition of the studied tailings was determined by X-ray analysis according to the standard method. X-ray analysis is a method of studying the matter structure by distribution in space and the intensity of X-ray radiation scattered on the analyzed object. A DRON-3 diffractometer was used for X-ray phase analysis (manufacturer—research and production enterprise Burevestnik, Russia). Recording signals in digital form allows for automatic data processing. Furthermore, the obtained data processing was manually carried out using a graphical editor or decrypted using a special program for the X-ray phase analysis of new crystalline formations. The operation of the graphic editor and the program used are described in detail in the study [36]. X-ray analysis was guided by accepted state standards [37,38].

Basic percolation leaching was carried out until the background grade was reached with the necessary time spent. The combination with activation was continued for 60 min.

Based on the series of experiments performed, it was determined that the activation of the tailings processing increases the recovery of valuable components from tailings.

A logarithmic dependence of leaching time and a polynomial regression of metal content in tailings was obtained from the experiment results.

A mathematical description of the processing by activation using flotation in a disintegrator plant were carried out. Mathematical studies were conducted in the form of active experiment planning with the selection of controllable factors (influences on the experimental results), which are demonstrated for specific deposits.

### 3. Results

*3.1. For Polymetallic Ores at the Sadonskoye Deposit (North Caucasus, Russia)*

Processing is carried out when working with heavy suspensions. The extraction of lead and zinc up to 80–85%, silver up to 60%, cadmium up to 56% and bismuth up to 30% is achieved, with tailings yields of 25–50%. The chemical composition of the tailings is characterized by the following proportion, %: $SiO_2$- 31; Fe—4.4; CaO—1.96; S—1; Ag—0.015; Cu—0.18; Mn—0.015; $K_2O$—3.5; $Al_2O_3$—0.8; $TiO_2$—0.03; Zn—0.95; Pb—0.84.

Volume of tailings in one trial—50 g.

The results of studies on the base leaching variant are shown in Tables 1 and 2 and Figure 5.

The regression equations showing the dependences of zinc and lead extraction are given in Table 2. The hypothesis of accepting the regression dependences as plausible was tested by the Fisher criterion with a reliability of 95% using the coefficients of determination.

Statistics on the leaching of processed ores in the disintegrator are shown in Tables 3 and 4 and Figure 6.

**Table 1.** Experimental plan for agitation leaching of tailings.

| No | Agent Content, g/L | | Ratio of Liquid to Solid | Agitation Leaching Duration, h | Extraction into Solution, % | |
|---|---|---|---|---|---|---|
| | $H_2SO_4$ | NaCl | | | Zn | Pb |
| 1 | 2 [−1] | 20 [−1] | 4 [−1] | 0.25 [−1] | 41.25 | 1.42 |
| 2 | 10 [1] | 20 [−1] | 4 [−1] | 0.25 [−1] | 57.75 | 0.49 |
| 3 | 2 [−1] | 160 [1] | 4 [−1] | 0.25 [−1] | 18.12 | 36.18 |
| 4 | 10 [1] | 160 [1] | 4 [−1] | 0.25 [−1] | 24.01 | 38.11 |
| 5 | 2 [−1] | 20 [−1] | 10 [1] | 0.25 [−1] | 48.43 | 3.58 |
| 6 | 10 [1] | 20 [−1] | 10 [1] | 0.25 [−1] | 82.12 | 4.77 |
| 7 | 2 [−1] | 160 [1] | 10 [1] | 0.25 [−1] | 12.64 | 30.94 |
| 8 | 10 [1] | 160 [1] | 10 [1] | 0.25 [−1] | 17.90 | 35.72 |
| 9 | 2 [−1] | 20 [−1] | 4 [−1] | 1 [1] | 44.59 | 17.15 |
| 10 | 10 [1] | 20 [−1] | 4 [−1] | 1 [1] | 70.27 | 1.42 |
| 11 | 2 [−1] | 160 [1] | 4 [−1] | 1 [1] | 10.94 | 24.77 |
| 12 | 10 [1] | 160 [1] | 4 [−1] | 1 [1] | 28.22 | 37.15 |
| 13 | 2 [−1] | 20 [−1] | 10 [1] | 1 [1] | 49.48 | 3.58 |
| 14 | 10 [1] | 20 [−1] | 10 [1] | 1 [1] | 50.54 | 1.78 |
| 15 | 2 [−1] | 160 [1] | 10 [1] | 1 [1] | 15.78 | 46.44 |
| 16 | 10 [1] | 160 [1] | 10 [1] | 1 [1] | 18.94 | 44.07 |
| 17 | 2 [−1] | 90 [0] | 7 [0] | 0.625 [0] | 21.38 | 35.82 |
| 18 | 10 [1] | 90 [0] | 7 [0] | 0.625 [0] | 34.64 | 41.68 |
| 19 | 6 [0] | 20 [−1] | 7[0] | 0.625 [0] | 67.78 | 1.65 |
| 20 | 6 [0] | 160 [1] | 7 [0] | 0.625 [0] | 25.78 | 45.01 |
| 21 | 6 [0] | 90 [0] | 4 [−1] | 0.625 [0] | 40.85 | 21.28 |
| 22 | 6 [0] | 90 [0] | 10 [1] | 0.625 [0] | 36.85 | 58.34 |
| 23 | 6 [0] | 90 [0] | 7 [0] | 0.25 [−1] | 40.54 | 49.18 |
| 24 | 6 [0] | 90 [0] | 7 [0] | 1 [1] | 42.75 | 40.84 |

In parentheses are the values for dummy independent variables: −1, 0, 1.

**Table 2.** Regression dependencies of zinc and lead extraction.

| Regression Equations | Significance Indicators |
|---|---|
| $\varepsilon_{Zn} = 39.35 + 6.76X_1 − 18.88X_2 − 0.62X_4 − 11.6X_1{}^2 + 7.19X_2{}^2 + 2.03X_4{}^2$ $-2.84X_1X_2 − 1.39X_1X_3 − 0.89X_1X_4 − 2.04X_2X_3 + 1.00X_2X_4 − 2.45X_3X_4$ | $R^2 = 0.9393$; $S_{ad} = 46.93$; F = 68.59 |
| $\varepsilon_{Pb} = 42.43 + 16.8X_2 + 2.68X_3 + 0.93X_4 − 3.89X_1{}^2 − 19.31X_2{}^2 + 2.36X_4{}^2$ $+ 2.12X_1X_2 − 0.9X_1X_4 + 1.73X_2X_3 + 1.04X_3X_4$ | $R^2 = 0.8888$; $S_{ad} = 71.17$; F = 30.19 |

The definition of dummy variables is carried out according to the formulas: $X_1 = \frac{C_{H_2SO_4}-6}{4}$; $X_2 = \frac{C_{NaCl}-90}{70}$; $X_3 = \frac{(\text{liquid to solid})-7}{3}$; $X_4 = \frac{t-0.625}{0.375}$.

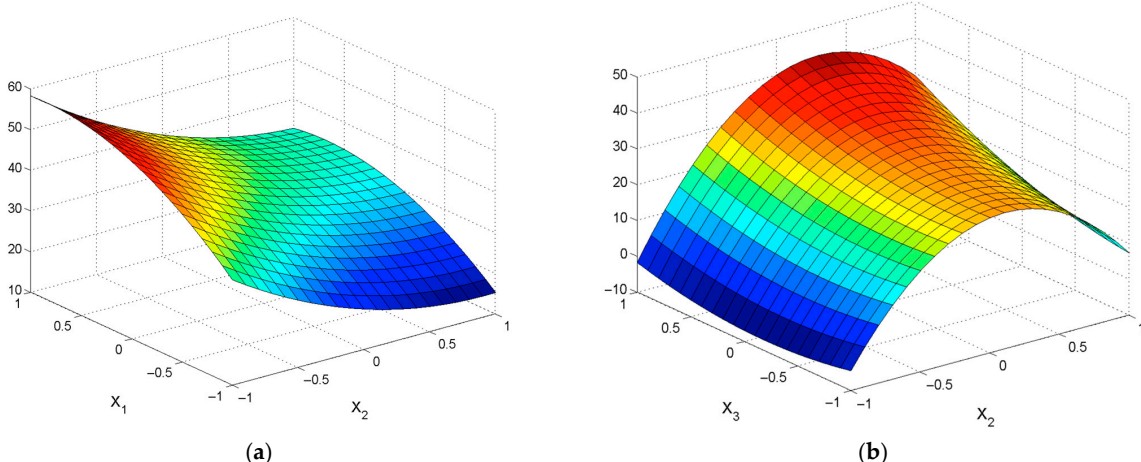

(a)  (b)

**Figure 5.** Dependency graphs for agitation leaching of tailings: (**a**) For zinc; (**b**) For lead.

**Table 3.** Experimental plan and statistics for leaching of tailings in the disintegrator.

| No | Agent Content, g/L | | Ratio of Liquid to Solid | Rotational Rate, Hz | Extraction into Solution, $\varepsilon$, % | |
|---|---|---|---|---|---|---|
| | $H_2SO_4$ | NaCl | | | Zn | Pb |
| 1 | 2 [−1] | 20 [−1] | 4 [−1] | 50 [−1] | 26.95 | 0.33 |
| 2 | 10 [1] | 20 [−1] | 4 [−1] | 50 [−1] | 78.74 | 0.95 |
| 3 | 2 [−1] | 160 [1] | 4 [−1] | 50 [−1] | 10.95 | 27.14 |
| 4 | 10 [1] | 160 [1] | 4 [−1] | 50 [−1] | 27.37 | 40.50 |
| 5 | 2 [−1] | 20 [−1] | 10 [1] | 50 [−1] | 47.37 | 4.76 |
| 6 | 10 [1] | 20 [−1] | 10 [1] | 50 [−1] | 54.74 | 1.79 |
| 7 | 2 [−1] | 160 [1] | 10 [1] | 50 [−1] | 6.32 | 40.48 |
| 8 | 10 [1] | 160 [1] | 10 [1] | 50 [−1] | 15.79 | 35.71 |
| 9 | 2 [−1] | 20 [−1] | 4 [−1] | 200 [1] | 32.42 | 0.71 |
| 10 | 10 [1] | 20 [−1] | 4 [−1] | 200 [1] | 61.47 | 1.43 |
| 11 | 2 [−1] | 160 [1] | 4 [−1] | 200 [1] | 13.47 | 27.14 |
| 12 | 10 [1] | 160 [1] | 4 [−1] | 200 [1] | 27.37 | 40.00 |
| 13 | 2 [−1] | 20 [−1] | 10 [1] | 200 [1] | 42.11 | 5.95 |
| 14 | 10 [1] | 20 [−1] | 10 [1] | 200 [1] | 52.63 | 1.55 |
| 15 | 2 [−1] | 160 (1) | 10 [1] | 200 [1] | 12.63 | 44.05 |
| 16 | 10 (1) | 160 (1) | 10 [1] | 200 [1] | 65.26 | 18.38 |
| 17 | 2 [−1] | 90 [0] | 7 [0] | 125 [0] | 22.11 | 39.17 |
| 18 | 10 [1] | 90 [0] | 7 [0] | 125 [0] | 36.11 | 28.33 |
| 19 | 6 [0] | 20 [−1] | 7 [0] | 125 [0] | 58.95 | 1.67 |
| 20 | 6 [0] | 160 [1] | 7 [0] | 125 [0] | 23.58 | 47.50 |
| 21 | 6 [0] | 90 [0] | 4 (−1) | 125 [0] | 35.37 | 34.29 |
| 22 | 6 [0] | 90 [0] | 10 [1] | 125 [0] | 29.47 | 42.86 |
| 23 | 6 [0] | 90 [0] | 7 [0] | 50 (−1) | 32.42 | 38.33 |
| 24 | 6 [0] | 90 [0] | 7 [0] | 200 (1) | 26.53 | 40.83 |

In parentheses are the values for dummy independent variables: −1, 0, 1.

**Table 4.** Regression dependencies based on the results of experiment on leaching of processed ores in the disintegrator.

| Regression Equations | Significance Indicators |
|---|---|
| $\varepsilon_{Zn} = 32.15 + 11.4X_1 - 14.04X_2 + 0.68X_3 + 1.85X_4 - 2.90X_1{}^2 + 9.25X_2{}^2 - 2.53X_4{}^2 - 0.39X_1X_2 - 1.95X_1X_3 + 1.32X_1X_4 + 1.47X_2X_3 + 4.84X_2X_4 + 3.61X_3X_4$ | $R^2 = 0.8277$; $S_{ad} = 143.62$; F = 18.06 |
| $\varepsilon_{Pb} = 39.44 - 1.17X_1 + 16.76X_2 + 1.28X_3 - 0.55X_4 - 5.64X_1{}^2 - 14.81X_2{}^2 - 0.86X_3{}^2 - 4.09X_1X_3 - 1.42X_1X_4 - 0.42X_2X_3 - 1.00X_2X_4 - 0.82X_3X_4$ | $R^2 = 0.9483$; $S_{ad} = 35.09$; F = 44.58 |

The definition of dummy variables is carried out according to the formulas: $X_1 = \frac{C_{H_2SO_4}-6}{4}$; $X_2 = \frac{C_{NaCl}-90}{70}$; $X_3 = \frac{(\text{liquid to solid})-7}{3}$; $X_4 = \frac{f-125}{75}$.

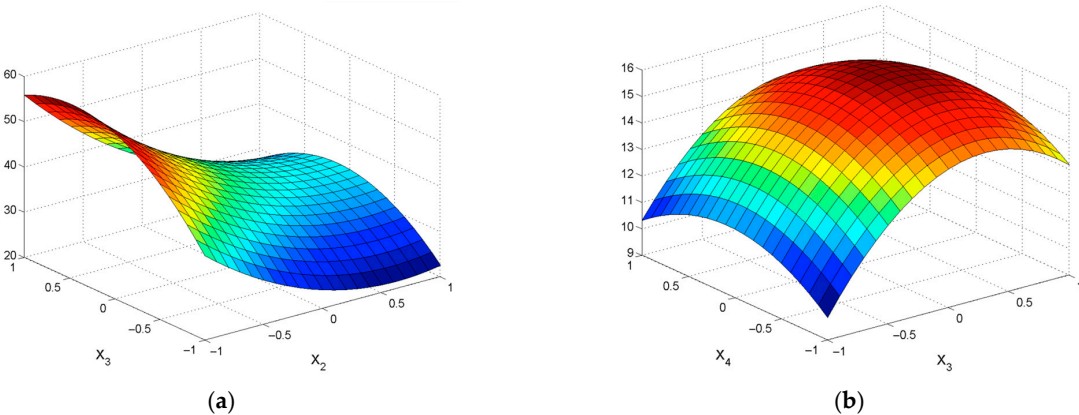

**Figure 6.** Dependency graphs for the leaching of processed ores in the disintegrator: (**a**) For zinc; (**b**) For lead.

The statistics on the multiple leaching of processed ore in the disintegrator are shown in Tables 5 and 6 and in Figure 7. With the sample volume $n$ = 120 in the experiment considering the independence and controllability of the input parameters, the model determines the resulting factors for the leaching of zinc and lead with a probability of 0.95.

**Table 5.** Experimental plan and statistics for multiple activation of tailings.

| No | Agent Content, g/L | | Rotational Rate, Hz | Number Of Activation Cycles | Extraction into Solution, $\varepsilon$,% | |
|---|---|---|---|---|---|---|
| | $H_2SO_4$ | NaCl | | | Zn | Pb |
| 1 | 2 [−1] | 20 [−1] | 50 [−1] | 7 [1] | 39.58 | 1.24 |
| 2 | 10 [1] | 20 [−1] | 50 [−1] | 3 [−1] | 64.42 | 1.19 |
| 3 | 2 [−1] | 160 [1] | 50 [−1] | 3 [−1] | 8.84 | 21.43 |
| 4 | 10 [1] | 160 [1] | 50 [−1] | 7 [1] | 26.52 | 40.48 |
| 5 | 2 [−1] | 20 [−1] | 50 [−1] | 3 [−1] | 53.68 | 3.57 |
| 6 | 10 [1] | 20 [−1] | 50 [−1] | 7 [1] | 70.53 | 2.38 |
| 7 | 2 [−1] | 160 [1] | 50 [−1] | 7 [1] | 1.05 | 39.29 |
| 8 | 10 [1] | 160 [1] | 50 [−1] | 3 [−1] | 17.89 | 45.00 |
| 9 | 2 [−1] | 20 [−1] | 200 [1] | 7 [1] | 40.58 | 0.95 |
| 10 | 10 [1] | 20 [−1] | 200 [1] | 3 [−1] | 52.26 | 0.33 |
| 11 | 2 [−1] | 160 [1] | 200 [1] | 3 [−1] | 10.11 | 20.95 |
| 12 | 10 [1] | 160 [1] | 200 [1] | 7 [1] | 29.05 | 48.10 |
| 13 | 2 [−1] | 20 [−1] | 200 [1] | 3 [−1] | 56.84 | 4.76 |
| 14 | 10 [1] | 20 [−1] | 200 [1] | 7 [1] | 66.32 | 2.38 |
| 15 | 2 [−1] | 160 [1] | 200 [1] | 7 [1] | 7.37 | 28.57 |
| 16 | 10 [1] | 160 [1] | 200 [1] | 3 [−1] | 22.11 | 48.81 |
| 17 | 2 [−1] | 90 [0] | 125 [0] | 5 [0] | 20.63 | 29.17 |
| 18 | 10 [1] | 90 [0] | 125 [0] | 5 [0] | 40.53 | 26.67 |
| 19 | 6 [0] | 20 [−1] | 125 [0] | 5 [0] | 63.37 | 2.50 |
| 20 | 6 [0] | 160 [1] | 125 [0] | 5 [0] | 24.32 | 28033 |
| 21 | 6 [0] | 90 [0] | 125 [0] | 5 [0] | 20.21 | 19.05 |
| 22 | 6 [0] | 90 [0] | 125 [0] | 5 [0] | 55.79 | 46.43 |
| 23 | 6 [0] | 90 [0] | 50 [−1] | 3 [−1] | 35.37 | 33.33 |
| 24 | 6 [0] | 90 [0] | 200 [1] | 7 [1] | 35.37 | 33.33 |

In parentheses are the values for dummy independent variables: −1, 0, 1.

**Table 6.** Regression dependencies based on the results of experiment on the multiple leaching of processed ore in the disintegrator.

| Regression Equations | Significance Indicators |
|---|---|
| $\varepsilon_{Zn}$ = 38.15 + 10.66$X_1$ − 15.17$X_2$ + 2.42$X_3$ − 1.37$X_4$ − 6.36$X_1{}^2$ + 3.92$X_2{}^2$ − 2.99$X_3{}^2$ − 1.68$X_4{}^2$ − 4.85$X_1X_2$ − 4.62$X_1X_3$ + 2.1$X_1X_4$ − 3.56$X_2X_3$ + 1.95$X_2X_4$ + 1.6$X_3X_4$ | $R^2$ = 0.9206; $S_{ad}$= 73.40; F = 30.72 |
| $\varepsilon_{Pb}$ = 40.94 + 16.12$X_2$ + 4.13$X_3$ + 0.67$X_4$ − 6.37$X_1{}^2$ − 17.44$X_2{}^2$ + 3.58$X_3{}^2$ + 1.36$X_4{}^2$ + 4.04$X_1X_2$ − 1.32$X_1X_3$ + 2.47$X_2X_3$ − 2.00$X_2X_4$ − 0.72$X_3X_4$ | $R^2$ = 0.9535; $S_{ad}$= 29.69; F = 55.26 |

The definition of dummy variables is carried out according to the formulas: $X_1 = \frac{C_{H_2SO_4}-6}{4}$; $X_2 = \frac{C_{NaCl}-90}{70}$; $X_3 = \frac{f-125}{75}$; $X_4 = \frac{K-5}{2}$.

The average statistics on the metal leaching of solution in the disintegrator: for lead is from 13 to 33%; for zinc is from 10 to 45% (Table 7).

A comparison was also made between the method of agitation for the leaching of zinc and lead with the method of activation in the working body of the disintegrator and leaching outside the working body. The results show relatively equal percentages of metal extraction but with a reduction in the processing time from a 15 to 16 min range to a tenth of a second, that is, by two orders of magnitude. Factors affecting metal extraction ranked by relevance: percentage of agent concentration in the leaching solution, rotational rate in disintegrator, ratio of liquid to solid.

**Table 7.** Parameters for the extraction of metals according to the method with pre-activation in the disintegrator.

| Extraction of Metals in Agitator | | | | Extraction in Disintegrator for 10 s | | | |
|---|---|---|---|---|---|---|---|
| concentration in tailings, % | | | | concentration in tailings, % | | | |
| zinc—0.94 | | lead—0.85 | | zinc—0.94 | | lead—0.85 | |
| Extraction for 0.2–1.0 h, % | | Extraction for 0.2–1.0 h, % | | Extraction for 10 s, % | | Extraction for 10 s, % | |
| extracted | left | extracted | left | extracted | left | extracted | left |
| 24 | 72 | 17 | 71 | 29 | 68 | 25 | 61 |

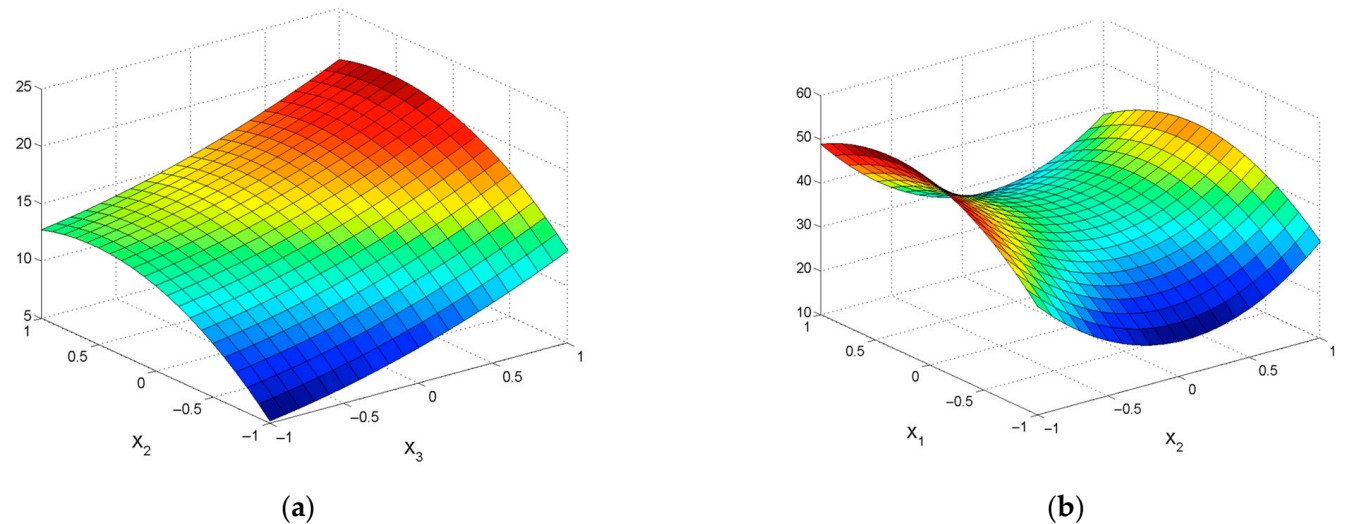

(a) (b)

**Figure 7.** Dependency graphs for multiple activation of tailings: (**a**) For zinc; (**b**) For lead.

*3.2. For Ferruginous Quartzites of the Kursk Magnetic Anomaly*

The processed ore with magnetic separations and composition of ferruginous quartzites is a finely dispersed mineral powder containing particles not exceeding 0.071 mm in size. The percentage of these particles is from 40 to 70 of the total mass.

Chemical composition of ore tailings: $SiO_2$—64%; Fe—8%; $Al_2O_3$—5.2%; Mn—3.2%; $K_2O$—0.7%; P—0.1%; Ca—0.8%; MgO—0.2%; Cu—$5 \cdot 10^{-3}$%; Ni—$4 \cdot 10^{-3}$%; Zn—$5 \cdot 10^{-4}$%; As, Ba, Be, Bi, Co, Cr, Li, Mo, Nb, Pb, Sn, Sr, Ti, V, Y—with a concentration of $(25–50) 10^{-5}$ %.

In one leaching operation, the share of iron extraction is 1.2% (Table 8).

**Table 8.** Extraction of iron from processed ores with ferruginous quartzites.

| Extraction in Activator | | Extraction in the Working Body of Disintegrator for 10 s, % | |
|---|---|---|---|
| concentration in tailings, % of iron—8 | | concentration in tailings, % of iron—8 | |
| Duration of extraction: 0.2 to1.0 h, % | | Duration of extraction: 10 s,% | |
| extracted | left | extracted | left |
| 0.8 | 7.36 | 1.2 | 7.4 |

The performance of the agitation leaching of processing tailings is shown in Tables 9 and 10 and Figures 8 and 9.

**Table 9.** Experimental plan and statistics for the agitation leaching of processed ore with ferruginous quartzites.

| No | Concentration in Solution, C, g/L | | Ratio of Liquid to Solid | Leaching Duration, t, h | Fe-Content in the Production Solution, $\varepsilon$, % |
|---|---|---|---|---|---|
| | $H_2SO_4$ | NaCl | | | |
| 1 | 2 [−1] | 20 [−1] | 4 [−1] | 0.25 [−1] | 0.40 |
| 2 | 10 [1] | 20 [−1] | 4 [−1] | 0.25 [−1] | 3.08 |
| 3 | 2 [−1] | 160 [1] | 4 [−1] | 0.25 [−1] | 1.56 |
| 4 | 10 [1] | 160 [1] | 4 [−1] | 0.25 [−1] | 3.76 |
| 5 | 2 [−1] | 20 [−1] | 10 [1] | 0.25 [−1] | 0.84 |
| 6 | 10 [1] | 20 [−1] | 10 [1] | 0.25 [−1] | 3.72 |
| 7 | 2 [−1] | 160 [1] | 10 [1] | 0.25 [−1] | 1.84 |
| 8 | 10 [1] | 160 [1] | 10 [1] | 0.25 [−1] | 4.35 |
| 9 | 2 [−1] | 20 [−1] | 4 [−1] | 1 [1] | 0.45 |
| 10 | 10 [1] | 20 [−1] | 4 [−1] | 1 [1] | 3.48 |
| 11 | 2 [−1] | 160 [1] | 4 [−1] | 1 [1] | 1.77 |
| 12 | 10 [1] | 160 [1] | 4 [−1] | 1 [1] | 4.18 |
| 13 | 2 [−1] | 20 [−1] | 10 [1] | 1 [1] | 1.17 |
| 14 | 10 [1] | 20 [−1] | 10 [1] | 1 [1] | 3.93 |
| 15 | 2 [−1] | 160 [1] | 10 [1] | 1 [1] | 2.55 |
| 16 | 10 [1] | 160 [1] | 10 [1] | 1 [1] | 4.75 |
| 17 | 2 [−1] | 90 [0] | 7 [0] | 0.625 [0] | 1.64 |
| 18 | 10 [1] | 90 [0] | 7 [0] | 0.625 [0] | 3.13 |
| 19 | 6 [0] | 20 [−1] | 7 [0] | 0.625 [0] | 2.27 |
| 20 | 6 [0] | 160 [1] | 7 [0] | 0.625 [0] | 2.58 |
| 21 | 6 [0] | 90 [0] | 4 [−1] | 0.625 [0] | 1.53 |
| 22 | 6 [0] | 90 [0] | 10 [1] | 0.625 [0] | 2.01 |
| 23 | 6 [0] | 90 [0] | 7 [0] | 0.25 [−1] | 2.10 |
| 24 | 6 [0] | 90 [0] | 7 [0] | 1 [1] | 2.54 |

In parentheses are the values for dummy independent variables: −1, 0, 1.

**Table 10.** Regression dependencies on iron extraction into solution, significance test.

| Regression Equations | Significance Indicators |
|---|---|
| $\varepsilon = 2.095 + 1.231X_1 + 0.444X_2 + 0.275X_3 + 0.176X_4 + 0.29X_1^2 + 0.33X_2^2 + 0.325X_3^2 + 0.225X_4^2 - 0.127X_1X_2 + 0.047X_2X_3 + 0.036X_3X_4$ | $R^2 = 0.977$; $S_{ad} = 0.0673$; F = 225.99 |

The definition of dummy variables is carried out according to the formulas: $X_1 = \frac{C_{H_2SO_4} - 6}{4}$; $X_2 = \frac{C_{NaCl} - 90}{70}$; $X_3 = \frac{(liquid\ to\ solid) - 7}{3}$; $X_4 = \frac{t - 0.625}{0.375}$.

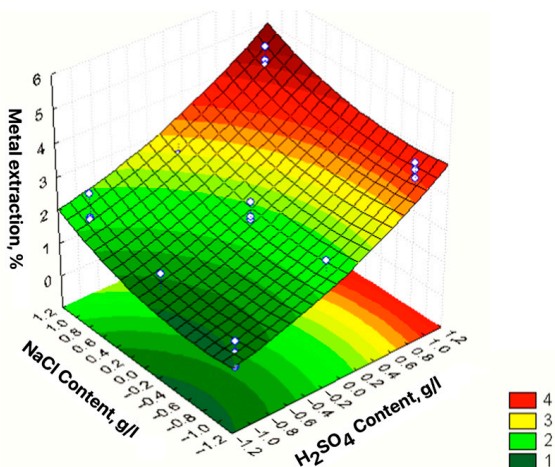

**Figure 8.** Graph of iron recovery, % of $H_2SO_4$ and NaCl concentration.

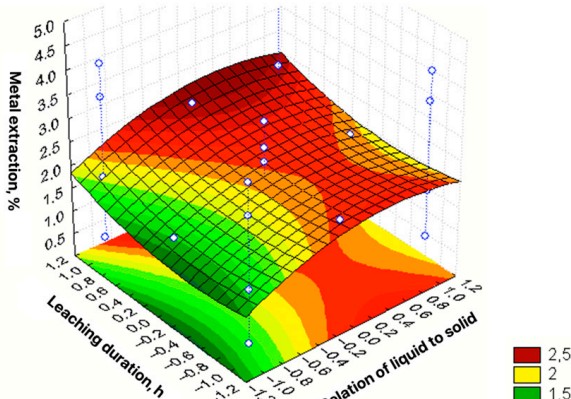

**Figure 9.** Graph of iron recovery, % of liquid to solid proportion and rotational rate in the disintegrator.

The performance of the leaching of processed tailings with ferruginous quartzites in the working body of disintegrator is shown in Tables 11 and 12 and Figures 10 and 11.

**Table 11.** Experimental plan and statistics for the leaching of tailings with ferruginous quartzites in the working body of the disintegrator.

| No | Concentration in Leaching Solution, C, g/L $H_2SO_4$ | NaCl | Rotational Rate, $X_4$, Hz | Leaching Duration, t, h | Fe-Content in the Production Solution, $\varepsilon$,% |
|----|----|----|----|----|----|
| 1 | 2 [−1] | 20 [−1] | 50 [−1] | 0.25 [−1] | 0.5 |
| 2 | 10 [1] | 20 [−1] | 50 [−1] | 0.25 [−1] | 5.33 |
| 3 | 2 [−1] | 160 [1] | 50 [−1] | 0.25 [−1] | 2.67 |
| 4 | 10 [1] | 160 [1] | 50 [−1] | 0.25 [−1] | 6.50 |
| 5 | 2 [−1] | 20 [−1] | 200 [1] | 0.25 [−1] | 1.45 |
| 6 | 10 [1] | 20 [−1] | 200 [1] | 0.25 [−1] | 6.44 |
| 7 | 2 [−1] | 160 [1] | 200 [1] | 0.25 [−1] | 3.18 |
| 8 | 10 [1] | 160 [1] | 200 [1] | 0.25 [−1] | 7.53 |
| 9 | 2 [−1] | 20 [−1] | 50 [−1] | 1 [1] | 0.78 |
| 10 | 10 [1] | 20 [−1] | 50 [−1] | 1 [1] | 6.02 |
| 11 | 2 [−1] | 160 [1] | 50 [−1] | 1 [1] | 3.06 |
| 12 | 10 [1] | 160 [1] | 50 [−1] | 1 [1] | 7.23 |
| 13 | 2 [−1] | 20 [−1] | 200 [1] | 1 [1] | 2.02 |
| 14 | 10 [1] | 20 [−1] | 200 [1] | 1 [1] | 6.80 |
| 15 | 2 [−1] | 160 [1] | 200 [1] | 1 [1] | 4.41 |
| 16 | 10 [1] | 160 [1] | 200 [1] | 1 [1] | 12.50 |
| 17 | 2 [−1] | 90 [0] | 125 [0] | 0.625 [0] | 2.84 |
| 18 | 10 [1] | 90 [0] | 125 [0] | 0.625 [0] | 5.41 |
| 19 | 6 [0] | 20 [−1] | 125 [0] | 0.625 [0] | 3.93 |
| 20 | 6 [0] | 160 [1] | 125 [0] | 0.625 [0] | 4.46 |
| 21 | 6 [0] | 90 [0] | 50 [−1] | 0.625 [0] | 2.65 |
| 22 | 6 [0] | 90 [0] | 200 [1] | 0.625 [0] | 2.98 |
| 23 | 6 [0] | 90 [0] | 125 [0] | 0.25 [−1] | 2.28 |
| 24 | 6 [0] | 90 [0] | 125 [0] | 1 [1] | 3.55 |

In parentheses are the values for dummy independent variables: −1, 0, 1.

**Table 12.** Regression dependencies on iron extraction in the disintegrator, significance test.

| Regression Equation | Significance Indicators |
|----|----|
| $\varepsilon = 3.091 + 2.381X_1 + 1.014X_2 + 0.698X_3 + 0.583X_4 + 1.035X_1^2 + 1.104X_2^2 - 0.276X_3^2 - 0.176X_4^2 + 0.259X_1X_3 - 0.268X_1X_4 + 0.255X_2X_3 + 0.339X_2X_4 + 0.315X_3X_4$ | $R^2 = 0.9384$; $S_{ad} = 1.0111$; $F = 43.03$ |

The definition of dummy variables is carried out according to the formulas: $X_1 = \frac{C_{H_2SO_4} - 6}{4}$; $X_2 = \frac{C_{NaCl} - 90}{70}$; $X_3 = \frac{f - 125}{75}$; $X_4 = \frac{t - 0.625}{0.375}$.

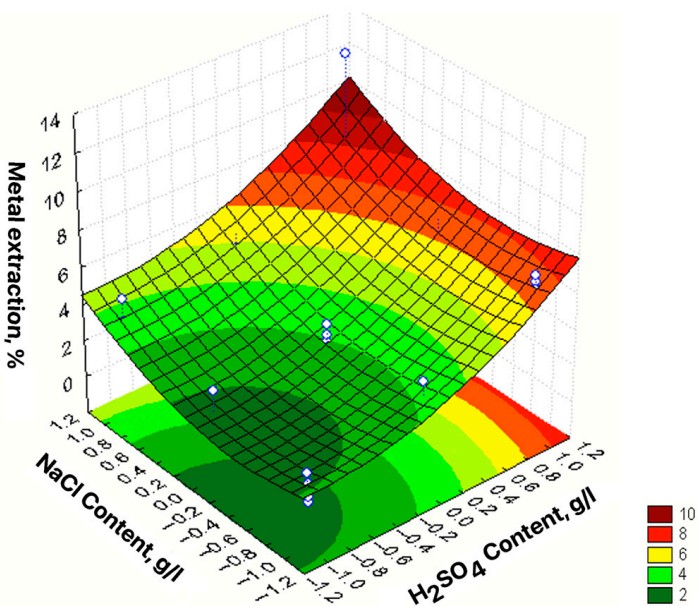

**Figure 10.** Graph of iron recovery, % depending on $H_2SO_4$ and NaCl shares.

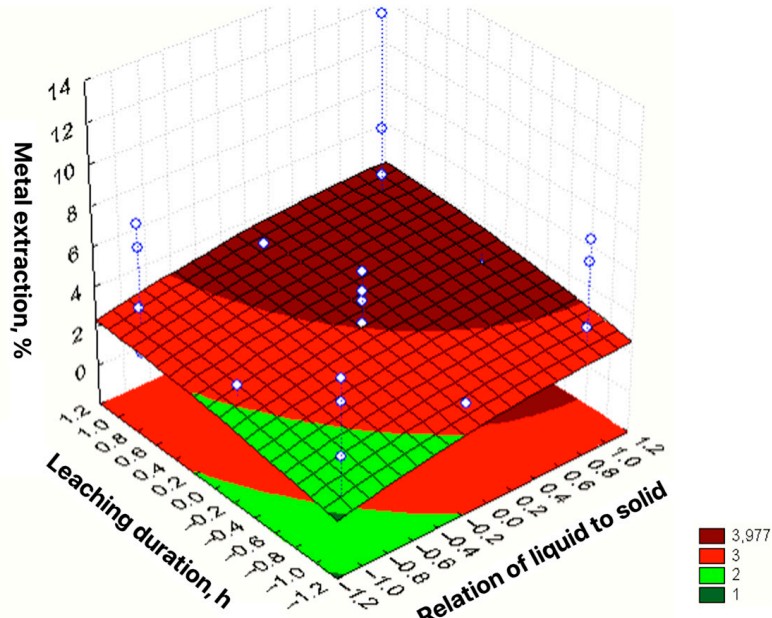

**Figure 11.** Graph of iron recovery, % depending on liquid to solid proportion and leaching duration in disintegrator.

Experiments have shown that the average percentage of iron in the processing tailings was 8 percent. A single pass of the material through the disintegrator allows one percent iron extraction, while a triple pass of the tailings through the disintegrator followed by leaching in the solution allows three percent iron extraction. The variants considered in the experimental plan yielded this proportion of iron leaching for only seven passes of material and subject to leaching in the disintegrator.

The leaching approach in the working body of the disintegrator compared to the activation method in the disintegrator and the leaching process outside the disintegrator shows almost the same percentage of iron recovery. However, the extraction effect with leaching outside the disintegrator is achieved in 15 to 60 min, while with activation in the integrator, iron recovery is achieved within 10 seconds, which is a hundred times faster.

The experimentally obtained data of the tables in all cases of the regression equation solution with appropriate determination coefficients based on Fisher's criterion at the significance level of 95% were accepted as plausible. This allows us to state with a probability of 95% that the method of mechanochemical activation is more effective in comparison with the traditional method.

The time of iron extraction from the tailings during mechanochemical activation in comparison with the basic method is reduced from 36 min to 10 s, which is 216 times. The share of extracted iron increases from 0.8 to 1.2, which is one and a half times. The results of the study indicate an increase in the rate of extraction of zinc and lead by 200 times during mechanochemical activation in comparison with the basic method.

Other factors affecting the leaching of valuable components include, in order of relevance, the level of agent concentration in the leaching solution, rotation rate in disintegrator and the ratio of liquid to solid.

On the basis of the studies carried out, it can be concluded that the leaching of tailings in the disintegrator working body, when compared to agitation leaching, increases the recovery rate of all studied valuable components in the solution from 10% to 25%. Due to the increased recovery rate of metals, secondary tailings can meet environmental regulations if a sufficient number of crushing cycles are performed. The metal concentration in the processing tailings for the elements As, Ba, Be, Bi, Co, Cr, Li, Mo, Nb, Pb, Sb, Sn, Sr, Ti, V and Y is $(30–50)\cdot10^{-6}$ %. The share of extraction of copper is $3.8–4.3\cdot10^{-3}$, nickel is $2.9–3.5\cdot10^{-3}$ and zinc is $3.2–4.2\cdot10^{-4}$, the same order of magnitude as for the other metals.

### 3.3. For Coal in the Regions of the Russian Federation

Resource-saving coal mining technologies that reduce operational losses confirm the relevance of the topic of mining waste processing [39]. Leaching of valuable components from coal mining wastes is possible, although, for traditional processing methods, the metal content is too low, and the process proves too costly.

Technologies with metal leaching and full recycling of coal waste will become economically viable if the risks of real damage to the ecological system and the environment due to the presence of potentially hazardous substances on the ground are considered.

The extraction from leaching mixtures into dry and quenched product are, %: for lead—43, for zinc—37, for chrome—18, for manganese—1.4 (Table 13) [4].

**Table 13.** Shares of the extraction of selected metals from tailings of coal processing.

| From the Tails in the Agitator for 0.2–1.0 h, % | | | | | From the Tails in Disintegrator for 10 s, % | | | | |
|---|---|---|---|---|---|---|---|---|---|
| Amount in tailings, g/t | | | | | Amount in tailings, g/t | | | | |
| Manganese | Nickel | Chrome | Lead | Zinc | Manganese | Nickel | Chrome | Lead | Zinc |
| 319 | 24 | 84 | 54 | 51 | 321 | 27 | 87 | 54 | 50 |
| Extraction into concentrate, % | | | | | Extraction into concentrate, % | | | | |
| 1.0 | 0.9 | 14 | 31 | 33 | 1.4 | 1.2 | 18 | 43 | 37 |

The experimental data, with an average sample value of $n = 100$, confirm the fundamental possibility of extracting metals from coal tailings by leaching with a high probability (up to 95%) [4,10].

### 3.4. Particle Size Distribution of Tailings

Tailings from the mechanochemical activation of waste are a dispersed mass, which consists of particles smaller in size than the original particles and more uniformly structured (Table 14).

A comparison of the basic agitation leaching of metals with preliminary mechanical activation in the disintegrator and subsequent leaching outside the disintegrator shows that, at the significance level of $\alpha = 0.05$, the proportion of lead recovery increased by 1.4 times and the proportion of zinc recovery increased by 1.1 times. Consequently, the share

of metals for secondary tailings processing under the new methodology proportionally decreases.

**Table 14.** The particle size distribution of tailings.

| Type of Waste | Residuals on Sieves, % | | | | |
|---|---|---|---|---|---|
| | 0.2 | 0.14 | 0.071 | <0.071 | Total |
| Iron quartzites of Lebedinsky Mining and Processing Plant | | | | | |
| original | 13.7 | 27.6 | 21.4 | 37.3 | 100 |
| after agitation leaching | 15.3 | 26.4 | 21.7 | 38.6 | 100 |
| after leaching in the disintegrator | 9.4 | 20.1 | 25.5 | 55.0 | 100 |
| Polymetallic ores of the Sadon lead-zinc Plant | | | | | |
| original | 18.6 | 29.3 | 12.9 | 39.2 | 100 |
| after agitation leaching | 16.7 | 28.0 | 13.3 | 42.0 | 100 |
| after leaching in the disintegrator | 13.4 | 24.1 | 20.3 | 52.2 | 100 |
| Hard coal of Russian regions | | | | | |
| original | 19.5 | 27.2 | 30.5 | 42.8 | 100 |
| after agitation leaching | 11.4 | 16.0 | 29.6 | 43.0 | 100 |
| after leaching in the disintegrator | 8.9 | 15.3 | 33.7 | 57.9 | 100 |

*3.5. Efficiency of Tailings Utilization*

Environmental safety must be understood as the implementation of the efficient disposal of industrial waste, including tailings. Such technology will minimize the impact on the environment.

Creating such technologies requires consideration of the cost of neutralizing environmental pollution with the simultaneous modeling of maximum permissible concentrations of pollutants.

The disposal of tailings from mineral processing can be effective depending on the type of application. A universal scheme on waste-free mining considers all stages of the process, from extraction to the use of the materials obtained from the processing tailings as a commodity.

An economic profit from the utilization of processing tailings is the result of the optimal solution in the mining-processing system based on V.A. Shestakov's mathematical model.

The evaluation of the effectiveness of tailings utilization contains the following steps:

- Study of the raw material base for the application of techniques for the neutralization of processed products;
- Analysis of market conditions in relation to new products;
- Study of the quality of traditional and newly developed products;
- Analysis of financial flows on means of production during tailings disposal;
- Comparison of the value of mineral raw materials and the products obtained from their processing.

The challenge of the economic approach to determining effectiveness is the need to consider the increased risk of non-waste disposal, including the effects of increased mining depth, environmental impacts, etc. The evaluation of options is carried out in the interrelationship of processes from field exploration to the sale of products.

The economic effect of introducing advanced technologies is the extraction of large volumes of valuable components from existing raw materials at comparable processing costs. The use of waste from raw material processing increases the volume of production, which affects the future fate of mineral resources (Figure 12).

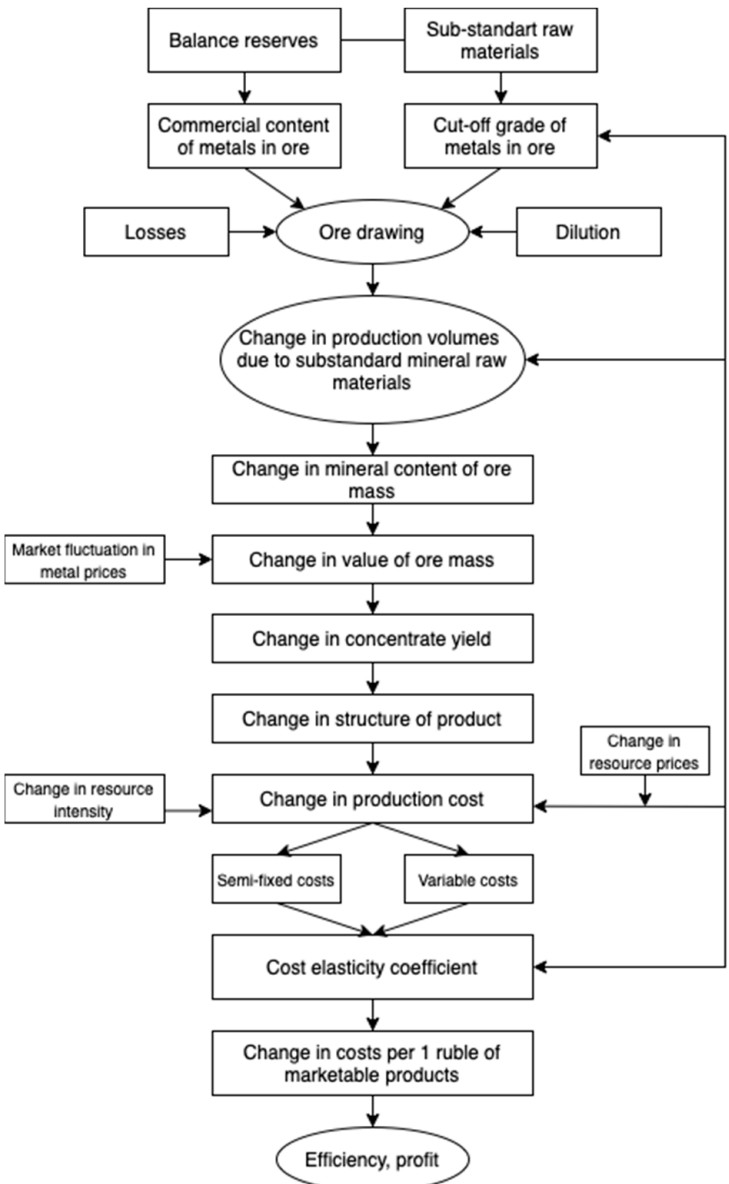

**Figure 12.** Algorithm for determining the efficiency of tailings utilization.

For most companies in the mining sector, the recycling of industrial waste to neutralize the leftover harmful components has environmental and economic benefits. This technology does not consider the profit from the extracted valuable components from industrial waste and the further use of treated industrial waste in industry. It is sufficient to consider the risks from the presence of hazardous components in industrial waste and the likelihood of their spread and impact. Such risks are noticeably reduced when the harmful effects are neutralized by reducing hazardous elements in industrial waste [7] or mined-out gas [40,41] to zero.

$$E = \sum_{t=1}^{T} \left( \sum_{i=1}^{n} C_{\sigma_{it}} - \sum_{i=1}^{n} C_{o_{it}} \right) Q_t,$$

where $C_{\sigma_{it}}$—direct maintenance costs of processed ore of $i$-type for $t$-period; $C_{o_{it}}$—indirect maintenance costs of processed ore of $i$-type for $t$-period; $Q_t$—volumes of neutralization tails for $t$-period; $n$—types of tailings, $i = 1, 2, \ldots n$.

Experiments have established that new technologies can increase the extraction of valuable components from substandard raw materials from 50% to 80%. It depends on the concentration of components in the processed tailings, which makes it possible to reduce

the residual proportion of metals in secondary tailings to the level of sanitary requirements. Such results have been obtained from studies of processing tailings of various types: on polymetallic ores from Sadon, on coals mined in Russian regions and on ferruginous quartzites from the Kursk Magnetic Anomaly. The similarity of the results of different regions confirms their correctness.

The exclusivity of goods obtained on the basis of mechanochemical technology lies in the prospect of using secondary processed ores as components of hardening mixtures. Such mixtures can be not only used for passive filling, but also for the preparation of binding solutions. With an increase in the binding forces of components by twenty percent, the strength of the solution at compression increases to 1 MPa [31,32].

The economic efficiency from the utilization of tailings can be achieved by replacing cement with a binder based on tailings. In the utilized tailings, 85% of particles are comparable to the cement particle size of 0.076 mm. This indicates the possibility of replacing cement with hardening mixtures without a large loss in strength. When the proportion of active particles is less than 85%, the strength of the mixture with an alternative binder is inferior to cement-based mixtures; when the proportion of particles is more than 85%, the strength of the mixture slightly increases, but remains comparable to the strength at 85%.

It should be noted that in the resulting liquid waste, residual solutions are used in the preparation of backfill instead of hazardous solid waste. In this case, the interaction of sulfides with acids produces soluble sulfates with the transfer of the valuable component (copper, zinc, iron, etc.) into a solution in the form of sulfate salts ($CuSO_4$, $ZnSO_4$, $FeSO_4$ etc.). When the solutions are neutralized, the metals precipitate, and sodium sulfate, which is a binder, remains in the solution.

The process of mechanoactivation makes it possible to practically extract all the valuable components present in substandard raw materials, reaching the level of sanitary requirements. This allows the secondary tailings to become suitable for creating hardening solutions, as well as for developing new commercial products [25,26].

The attraction of large reserves of mineral resources in the production makes it possible to create a new raw material base in the mining and processing industry. In addition, it saves resources in terms of new deposits, which is of particular relevance due to the shortage of certain metals, helpful in the creation of an additional mineral resource base in Russia [27].

Conclusions regarding the profitability of the method can be made considering the types of metals, ore development conditions, technological qualities, etc. The profit function depends on many factors, which require detailed study.

For example, the mining company ERGO is one of the largest enterprises in the world that processes the stale tailings of gold- and uranium-bearing dumps. The raw material base of the enterprise is cyanidation tailings from eighteen old storage facilities in the East Rand area. The total reserves are up to 400 million tons, and the average gold content is 0.47 g/t, for uranium—34 g/t, while the mass fraction of pyrite sulfur is 0.63% [25,26]. The Norilsk Mining and Metallurgical Plant produces a product with a content of platinum of up to 100 g/t, palladium—20 g/t, gold—25 g/t, silver—90 g/t from tailings of the processing plant. The Bashkir Copper and Sulfur Combine produces a concentrate with a gold content of 37.6 g/t and silver of 35.6 g/t from the tailings of the processing plant. The Tajikzoloto Association produces a concentrate with a gold content of 77 g/t from tailings.

## 4. Conclusions

World practice shows that the real criterion for reducing the risks of nature pollution from tailings is the complete neutralization of chemically hazardous tailings. Such an effect can be achieved by transferring the tailings after mechanical activation into the leaching solution in the disintegrator. This research confirmed the following findings:

- Leaching in a disintegrator in comparison with other options provides an approximately equal extraction of metals, but it is faster by two orders of magnitude (from 15–60 min to several seconds).
- The leaching of metals in a disintegrator increases the recovery by 10–25% compared to the agitation leaching method. Increasing the cycles of processing will make it possible to achieve the environmentally safe presence of metals in tailings.
- Factors influencing the process in decreasing order of influence: the proportion of agent in the leaching solution, the disintegrator rotation frequency and the ratio of liquid to solid in the slurry.
- The economic effect of mechanoactivation in a disintegrator is a greater extraction of metals from mineral raw materials at almost the same cost. This effect is defined as the difference between the damage from environmental pollution and the cost of tailings neutralization. An assessment of the threat from tailings in the natural environment shows that the technology of tailings neutralization provides a profit to the national economy, even if commercial products are not produced from the processing of ores.
- Methods of mechanochemical activation of tailings make it possible to extract a sufficient proportion of metals in accordance with environmental standards. This allows us to use processed materials without limitations, and therefore, to present mechanoactivation as an almost waste-free technology.
- The conditions for the applicability of a compromise optimal criterion for the extraction of metals from tailings are formulated on the basis of mathematical experiment planning, methods of regression analysis and considering sanitary standards for the neutralization of tailings.

The use of existing huge reserves of mineral resources in production creates a fundamentally new approach for the mining industry. This will expand the list of rare earth metals to be mined, which is relevant for the creation of an additional mineral resource base for individual enterprises and Russia as a whole.

The following provisions need to be implemented in order for waste-free technology to be effective:

- Tailings after the extraction of valuable components from them will become raw materials for new products;
- The use of tailings as hardening mixtures improves the properties of materials and reduces the costs of ore processing;
- The use of mechanical activation will contribute to the development of the mining industry, as strategically important mineral ores will be rationally used.

The approaches proposed in the new technology must be adapted to the modern conditions of the mining industry.

**Author Contributions:** Conceptualization, V.I.G. and M.M.K.; methodology, M.F.M.; software and validation, Y.V.A. and E.E.A.; formal analysis and investigation, N.V.R.-L., G.V.K. and O.A.K.; resources, E.V.T. and O.O.S.; data curation, Y.V.A. and E.E.A.; writing—original draft preparation, V.I.G.; writing—review and editing, V.I.G. and M.F.M.; visualization, E.V.T.; supervision and project administration, M.M.K. All authors have read and agreed to the published version of the manuscript.

**Funding:** This research received no external funding.

**Data Availability Statement:** No new data were created or analyzed in this study. Data sharing is not applicable to this article.

**Conflicts of Interest:** The authors declare no conflict of interest.

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
