# Peer review of "Comprehensive Recovery of Metals in Tailings Utilization with Mechanochemical Activation"

_resources, doi:10.3390/resources12100113_

Round 1

Reviewer 1 Report

Dear authors,

Despite the large amount of experimental results, they are difficult to understand. You should rewrite the manuscript according to the requirements for experimental articles. Specifically, you should focus on the following aspects:

1.      Shorten the introduction: Provide a concise introduction that clearly outlines the purpose of the article.

2.      Clearly indicate the purpose of the article: State the main objectives of your research to guide readers in understanding the significance of your study.

3.      Describe in detail the methodology of experiments: Provide a comprehensive explanation of the experimental procedures and methods employed in your research.

4.      Present the results obtained: Clearly and succinctly present the findings of your experiments, using appropriate tables, graphs, or figures where necessary to enhance clarity.

5.      Draw conclusions: Based on the results, draw logical conclusions that directly address the stated objectives of the study.

Reviewer 2 Report

The paper presented by the authors suggests a new method of utilizing leaching technology by introducing a disintegrator to achieve faster results. The paper is well-written and backed by substantial experimental data and observations. The proposed method is intriguing because it is both technologically simple to implement, using existing technology, yet also offers promising results.

I do recommend the paper for publication, with a few minor questions to be commented/answered:

Minor questions:

Equation 1: I understand that the proposed equation is mainly for representative means, but since it is stated as a one-dimensional task, a question appears on the validity of this simplification. Since the proposed system of the disintegrator in  Figure 3 is spiral-like 2D, and most close to the problem would be a choice of a two-dimensional Cauchy problem in polar coordinates rather than one-dimensional. Thow, I understand that even a 1D approach can be good enough here; I would like to see some comments on this simplification in the paper.

Line 546: The statement: “Economic efficiency from the utilization of tailings can be achieved by replacing cement 546 with hardening mixtures because their particle size exceeds 0.076 mm, and the volume of 547 such fractions reaches up to 85%” is not entirely clear why such particle size would be a critical threshold? And we know that, in reality, any particle size is a distribution, would the statistical parameters of the distribution matter for such a conclusion? I would propose either deleting or rephrasing the statement for clarity.

Line 573: In conclusion, there is a statement regarding the economic considerations and benefits of the proposed method. Though the scientific novelty and potential profitability of the method are not questionable, I think, it is still too early to make a strong conclusion regarding the overall profitability of the method (authors can only propose such profitability). To make a statement, more deep analysis is to be made, including hardware and machinery usage, if the method is implemented, whether there would be an impact on lifetime, etc. I would suggest softening the statement or rephrasing it, indicating potential profitability.

Reviewer 3 Report

1. More in-depth analyses should be made after presenting the experimental results.

2. It is highly recommended to present key quantitative conclusions in the subsection of Conclusions.

3. Novelty and new findings from the current work must be outlined explicitly.

Minor editing of English language required.

Reviewer 4 Report

(1) Why use the formula in Table 2 for fitting? This formula is too complex.

(2) In Tables 3 and 4, the relevant dimensions should be indicated.

(3) In terms of comprehensive metal recovery, many scholars have conducted a lot of research, and some relevant literature is recommended to be cited:

Sb release characteristics of the solid waste produced in antimony mining smelting process

Antimony Ore Tailings: Heavy Metals, Chemical Speciation, and Leaching Characteristics

Evaluation of heavy metals stability and phosphate mobility in the remediation of sediment by calcium nitrate

Leaching and Releasing Characteristics and Regularities of Sb and As from Antimony Mining Waste Rocks

Factors on the distribution, migration, and leaching of potential toxic metals in the soil and risk assessment around the zinc smelter

Minor editing of English language required

Round 2

Reviewer 1 Report

Dear authors,

Your manuscript has been significantly improved after revisions; however, I still believe that it is not ready for publication. To proceed with manuscript, it is necessary to extensively revise Section 2. You should relocate the chemical compositions of the researched materials to this section. Additionally, apart from the chemical composition, it is desirable to include other characteristics, such as phase composition, particle size, etc. It is essential to provide the methods you employed to determine the chemical composition of the tailings and the obtained solutions or solid residues (method, equipment, make, and country of manufacture). Furthermore, it is crucial to specify the formulas used for calculating the extraction coefficients of the elements.

It would be highly beneficial to present the chemical and phase compositions of the leaching residues obtained from the most successful experiments. In your manuscript, you assert that the utilization of your technology reduces the environmental hazards of tailings storage and enables the recovery of valuable components from them. However, there is no description of how you propose to use the obtained solutions after leaching. Given the low concentrations of valuable elements in the tailings, the content of these elements in the obtained solutions is likely to be minimal and extracting them from the solutions might not be economically justified.

In this case, instead of hazardous solid waste, we would be dealing with hazardous liquid waste, the storage of which demands higher costs compared to solids. How do you suggest addressing this issue?

Additional comments:

Line 105-106:

Figure 1 is repeated

Line 179:

The numbers in the figure not clear.

Line 303:

“…For each type of tailings 0.05 t…” 0.05 t?

Line 324-333:

The text is repeated.

Line 516:

The equation is repeated.

Reviewer 4 Report

the paper can be considered to accept

the paper can be considered to accept

Author Response

Dear Reviewer,

My co-authors and I are very grateful to Reviewer for taking the time to read our manuscript again, for seeing the improvements and for the positive evaluation of the manuscript.

We edited the manuscript and improved English language.

Sincerely,

Authors

Round 3

Reviewer 1 Report

Dear authors,

In my opinion, the manuscript can be published after minor revision.

I have only one comment:

Figure 1 and Figure in lines 109-110 are identical.

Author Response

Dear Reviewer,

Thank you for your review and for your hard work. We apologize for this mistake in the manuscript. We removed the duplicate illustration of Figure 1.

Sincerely,

Team of authors